# Seasonal Dynamics of Ecosystem Carbon Exchange and Their Influencing Factors in Grasslands of Different Degrees of Salinization in Northern China

**DOI:** 10.3390/plants14182854

**Published:** 2025-09-12

**Authors:** Gaoliang Pang, Jingjing Wang, Jianyu Wang, Yicong Chen, Kuanhu Dong, Huajie Diao

**Affiliations:** 1College of Grassland Science, Shanxi Agricultural University, Jinzhong 030801, China; glpang2023@163.com (G.P.); 19324641285@189.cn (J.W.); wjy13196699795@163.com (J.W.); cyc02052024@163.com (Y.C.); dongkuanhu@sxau.edu.cn (K.D.); 2Shanxi Key Laboratory of Grassland Ecological Protection and Native Grass Germplasm Innovation, Jinzhong 030801, China; 3Youyu Loess Plateau Grassland Ecosystem Research Station, Shanxi Agricultural University, Shuozhou 037200, China

**Keywords:** saline–alkaline grasslands, carbon exchange, root-to-shoot ratio

## Abstract

Soil pH plays a critical role in shaping the structural composition and functional dynamics of grassland ecosystems. The seasonal dynamics of carbon exchange and the factors influencing them in grassland ecosystems along saline–alkaline gradients remain unclear. In this study, saline–alkaline grasslands in northern China were classified into four gradients based on soil pH: mild salinization (pH = 8.36 ± 0.01), moderate salinization (pH = 9.21 ± 0.06), severe salinization (pH = 9.92 ± 0.04), and extreme salinization (pH = 10.49 ± 0.01). Ecosystem carbon exchange (net ecosystem carbon exchange (NEE), ecosystem respiration (ER), and gross ecosystem productivity (GEP)), as well as related biotic and abiotic factors, were investigated in the years 2023 and 2024. Results indicated that extreme salinization significantly reduced NEE, ER, and GEP, whereas no significant differences were observed in these carbon flux components between moderate and severe salinization levels. In 2024, NEE, ER, and GEP exhibited seasonal dynamics; compared to the early (May) and late (September) periods, greater differences were observed during the middle (June–August) period, particularly across varying salinization gradients. Significant negative correlations were observed between soil temperature, root-to-shoot ratio (R/S) and NEE, ER, and GEP, while above-ground and below-ground biomass were significantly positively correlated with NEE, ER, and GEP. Soil moisture exhibited a significant quadratic relationship with both ER and GEP. Importantly, results showed that the R/S explained the greatest variation in carbon fluxes. In summary, as salinization increased, carbon exchange capacity declined significantly, particularly under conditions of extreme salinization, where the R/S emerged as the primary regulatory factor.

## 1. Introduction

Saline–alkaline soils are widely distributed throughout the world as one of the factors affecting ecological balance [1,2]. Saline–alkaline grasslands, as an important component of global saline–alkaline soils, are expanding in area under the combined effects of climate change and anthropogenic activities [3], and this expansion has emerged as a key factor that disrupts the balance of grassland ecosystems and hinders their sustainable development. Saline–alkaline soils are characterized by cation enrichment and high pH, which significantly alter the soil’s physicochemical properties [4]. Thus, saline–alkaline stress not only affects the efficiency of plant nutrient uptake and utilization, but also leads to changes in plant physiological functions, as well as in plant biomass and community structure [5,6]. The inhibition of plant growth by these stresses consequently affects grassland carbon exchange (net ecosystem carbon exchange (NEE), ecosystem respiration (ER) and gross ecosystem productivity (GEP)) processes [7]. Therefore, an investigation into the effects of salinization gradients in grasslands on carbon exchange will contribute to our understanding of carbon cycling, provide a foundation for ecological compensation, and facilitate a synergistic approach to restoring degraded grasslands while enhancing their carbon sequestration potential.

On natural grasslands, previous studies have shown that nutrient supplementation promotes plant growth, thereby increasing NEE, ER, and GEP [8]. Additional studies have shown that soil temperature and moisture increase NEE, ER, and GEP within a certain range [9]. Unlike natural grasslands, saline–alkaline grassland soils have higher pH levels, reducing the nutrients available to plants and hindering their growth [4,10]. Other studies on saline–alkaline grasslands have shown that high soil pH reduces plant biomass, leading to decreased ER, and GEP [11]. Saline–alkaline grasslands indirectly constrain NEE by decreasing resource use efficiency, a process driven by elevated soil pH [12]. As degraded grasslands, saline–alkaline grasslands exhibit altered community structure, which thereby hinders photosynthetic efficiency at the community level and leads to a reduction in GEP [13,14]. Increasing soil pH reduces heterotrophic respiration dominated by soil microorganisms, thereby decreasing ecosystem respiration [15,16]. Salinization inhibits plant root growth, which consequently reduces root autotrophic respiration and thereby contributes to a decline in ecosystem respiration [17,18].

As soil pH increases, plants exhibit species-specific adaptive responses by altering biomass and coverage [19]. With increasing salinization, several studies have found that both above-ground and below-ground biomass exhibit a decreasing trend, whereas the root-to-shoot ratio (R/S) increases to enhance adaptation to saline–alkaline stress [20]. Additionally, with increasing soil pH, species composition and community structure shift towards more saline-tolerant taxa as an adaptation to the effects of salinization [19,21]. Under mild saline–alkaline stress, plants enhance their leaf photosynthetic capacity through increased photosynthetic parameters, thereby constituting a key adaptive mechanism [22]. However, when stress exceeds the adaptive range, it impairs physiological function and disrupts community integrity, leading to widespread decline [6,23].

The agro-pastoral zone in northern China experiences varying degrees of salinization due to both natural and human factors, impacting plant productivity and grassland ecological functions. Therefore, we selected grassland patches with different degrees of salinization (mild, moderate, severe, and extreme) in the region. By measuring net ecosystem carbon exchange, ecosystem respiration, gross ecosystem productivity, and related biotic and abiotic factors, we investigated the characteristics and driving mechanisms of ecosystem carbon exchange in grasslands with varying degrees of salinization. Specifically, the following two major questions were addressed: (1) How does ecosystem carbon exchange and its seasonal dynamics manifest in grasslands ecosystems with different degrees of salinization?; (2) What are the regulatory mechanisms and main factors of ecosystem carbon exchange in grasslands with different degrees of salinization?

## 2. Results

### 2.1. Effects of Different Degrees of Salinization on Soil Physicochemical Properties and Plant Productivity

The maximum precipitation and average maximum temperature in 2023 both occurred in July. The maximum precipitation in 2024 occurred in June, while the average maximum temperature was recorded in July (Figure 1). Year, salinization, and their interaction significantly affected soil moisture (SM) (*p* < 0.01, Table 1). Soil moisture peaked in August in 2023 (Figure 2a), while SM exhibited unique seasonal dynamics in 2024; these dynamics followed a unimodal pattern that still reached its peak in August (Figure 2b). Soil moisture increased significantly with the degree of salinization across the study region. There were significant differences in soil moisture with different degrees of salinization. Compared to mild conditions, SM increased by 51.1%, 136.1%, and 242.6% under moderate, severe, and extreme conditions, respectively (Figure 2c; *p* < 0.05). Year, salinization, and their interaction significantly affected soil temperature (ST) (*p* < 0.05, Table 1). Soil temperature peaked in August 2023 (Figure 2d). In contrast, ST in 2024 exhibited distinct seasonal dynamics: it rose to a peak during the peak growth period (July–August) before declining (Figure 2e). Compared with mild conditions, severe and extreme conditions increased ST by 4.9% and 26.2%, respectively (Figure 2f).

Salinization significantly affected above-ground biomass (AGB), below-ground biomass (BGB), and R/S (*p* < 0.01). Specifically, both AGB and BGB decreased with intensifying salinization (Figure 3a,b). Relative to mild conditions, AGB was reduced by 33.2%, 36.9%, and 77.9% under moderate, severe, and extreme conditions, respectively; a more modest decline was observed for BGB, which decreased by 14.3%, 14.3%, and 28.9% under the same salinization gradients. No significant differences were observed between moderate and severe salinization for either AGB or BGB (Figure 3, *p* > 0.05). The R/S increased significantly with increasing salinization intensity. Relative to mild conditions, R/S was elevated by 35.4% under severe conditions and by 224.5% under extreme conditions, with the latter exhibiting a far more pronounced increase in magnitude (Figure 3c).

### 2.2. Effects of Different Degrees of Salinization on Ecosystem Carbon Exchange and Resource Use Efficiency

While year and salinization significantly affected NEE (*p* < 0.05) without a significant interaction, both factors along with their interaction had highly significant effects on ER and GEP (*p* < 0.01, Table 1). In 2023, NEE peaked in August, whereas ER and GEP both attained their peaks in July (Figure 4a,d,g). In 2024, NEE showed a V-shaped seasonal pattern with a peak in June (Figure 4b), whereas ER and GEP both followed a unimodal pattern, culminating in an August peak (Figure 4e,h). The differences in NEE, ER, and GEP across salinization gradients were most pronounced during the mid-growing season (June–August) under extreme conditions, compared to the early (May) and late (September) periods of the 2024 growing season. Saline–alkaline grasslands showed carbon sink potential, and NEE weakened with the intensification of salinization: compared with mild conditions, NEE decreased by 19.9%, 21.4%, and 69.7% under moderate, severe, and extreme conditions, respectively (Figure 4c). Relative to mild conditions, ER declined by 9.9%, 8.9%, and 56.1%, while GEP decreased by 12.2%, 11.8%, and 59.3% across the salinization gradient (Figure 4f,i). No significant difference in NEE, ER, or GEP was observed between moderate and severe salinization conditions (Figure 4, *p* > 0.05).

Year had a significant effect on CUE (*p* < 0.05, Table 1). Salinization had significant effect on WUE (*p* < 0.01, Table 1). In both 2023 and 2024, there were no significant differences in CUE between grasslands with varying degrees of salinization (Figure 5a,b, *p* > 0.05). The two-year average CUE exhibited a significant decreasing trend with increasing salinization, declining by 24.6% under extreme compared to mild conditions (Figure 5c). Across the two-year, WUE exhibited a consistent and significant decreasing trend along the saline–alkaline gradients (*p* < 0.05). Compared with mild saline–alkaline conditions, the two-year average WUE decreased significantly by 31.0%, 47.6%, and 74.1% under moderate, severe, and extreme saline–alkaline conditions, respectively (Figure 5f).

### 2.3. Factors Influencing Ecosystem Carbon Exchange

Net ecosystem exchange (NEE) was negatively correlated with ST; similarly, ER and GEP also exhibited a negative correlation with ST (*p* < 0.01, Figure 6a). SM correlated negatively with NEE and quadratically with ER and GEP (*p* < 0.01; Figure 6b). The response of plant growth to SM was non-linear—initially promotive under moderate conditions within a certain range, but becoming inhibitory once SM surpassed a critical threshold. When SM was 16.23%, ER reached its maximum value of 8.13 μmol·m^−2^·s^−1^, and when SM was 15.1%, GEP reached its maximum value of 10.46 μmol·m^−2^·s^−1^. As soil pH increased, ER and GEP first decreased substantially, then flattened, and finally decreased rapidly; NEE was significantly and strongly negatively correlated with soil pH (*p* < 0.01, Figure 6c). AGB and BGB were positively correlated with NEE, ER, and GEP (*p* < 0.01, Figure 7a,b), while R/S was negatively correlated with these parameters (*p* < 0.01, Figure 7c). Under extreme salinization, the R/S significantly increased; GEP and ER were negatively correlated with R/S. Redundancy analysis showed that the factors explained 91.05% of ecosystem carbon exchange (Figure 8a), with R/S having the highest explanatory power for carbon exchange (Figure 8b).

## 3. Discussion

### 3.1. Effects of Different Degrees of Salinization on Carbon Exchange in Grassland Ecosystems

Results showed that grassland salinization significantly reduced NEE, ER, and GEP (Figure 4). High soil pH decreases soil available nutrients [24], which is detrimental to plant growth and thereby indirectly reduces carbon fluxes [12]. Another study has found that high salt ions in soils made the soil osmotic pressure higher than that of plant roots, resulting in physiological drought in plants and thereby hindering the uptake and transfer of water and nutrients, which ultimately inhibited plant growth [25]. In addition, salinization reduces below-ground biomass and consequently reduces root-dominated autotrophic respiration, thereby lowering carbon fluxes [18,26]. At the same time, higher soil pH inhibits above-ground plant growth, which reduces light energy capture by leaves and decreases above-ground plant-dominated GEP [11,27]. Furthermore, osmotic stress from saline–alkaline ions reduces stomatal aperture, leading to a disruption of plant-atmosphere gas exchange and a consequent decline in leaf photosynthetic and respiratory processes [28,29]. Results showed that below-ground biomass decreased significantly in saline–alkaline grasslands (Figure 3). The main reason is that high soil pH inhibits plant root growth; since plant roots have a close mutualistic relationship with soil microorganisms, this inhibition weakens microbial physiological functions [30,31], which indirectly affects carbon exchange. NEE showed a decreasing trend with intensifying salinization (Figure 4c). The direct manifestation of carbon sink potential is NEE. Compared with natural grasslands, saline–alkaline grasslands have a significantly weaker overall carbon sink potential [11]. The core mechanism is identified as follows: intensified salinization significantly reduces above-ground biomass, and the inhibition of plant photosynthesis and growth processes leads to a reduction in leaf-dominated GEP that exceeds the reduction in ER, ultimately weakening ecosystem carbon fixation capacity [7,12]. In summary, NEE, ER, and GEP decrease with increasing salinization.

Results showed that NEE, ER, and GEP decreased significantly under extreme salinization, while no statistically significant differences were detected under moderate and severe salinization conditions—this finding was linked to plant adaptive responses. In grasslands degraded by salinization, plants stabilize vegetation biomass by adjusting species composition and optimizing coverage, while further enhancing photosynthesis levels to build adaptive mechanisms against salinization stress [19,23]. In response to saline–alkaline stress, plants elevated their root-to-shoot ratio by increasing root proportion in our study. However, under extreme conditions, the saline–alkaline concentration exceeded plant tolerance thresholds, significantly reducing ecosystem carbon fluxes. This adaptive change enhances water and nutrient uptake efficiency, thereby improving stress tolerance and ultimately supporting better growth performance in plants [22]. Once environmental conditions surpass a critical threshold, root systems can boost their ability to compete for resources through morphological adjustments. Root length and root surface area, which are critical factors determining the root system’s water and nutrient uptake capacity, together play a role in enhancing stress resistance [32,33]. Furthermore, results showed that biomass showed no significant change under moderate and severe conditions, and that plants enhanced photosynthesis by increasing leaf area (despite no significant change in biomass) to adapt to saline–alkaline stress [34]. In conclusion, plant adaptability and biomass allocation strategies collectively regulate NEE, ER, and GEP dynamics in saline–alkaline grasslands.

### 3.2. Regulation of Carbon Exchange in Grassland Ecosystems

In our study, NEE, ER, and GEP were found to be significantly negatively correlated with ST across different degrees of salinization (Figure 6a). Previous studies have found that increased soil temperature promotes plant growth and carbon fluxes [35,36], but other studies have found that increased soil temperature can exacerbate soil salinization [37]. When ST increases, intensified evaporation leads to significant salt accumulation at the soil surface, thereby reducing species diversity and destabilizing plant communities [38], and these effects consequently exert negative impacts on carbon exchange. Our study further revealed that SM differentially regulated carbon exchange processes across saline–alkaline gradients. ER and GEP displayed unimodal responses to SM variation, peaking at intermediate moisture levels before declining thereafter, whereas NEE exhibited a consistent negative correlation with SM (Figure 6b). These patterns are attributable to increased SM, which enhances plant nutrient uptake [39], thereby stimulating plant growth and promoting ecosystem carbon exchange. However, once SM exceeds a critical threshold, the resulting declines in soil permeability and oxygen availability suppress aerobic microbial and root activity [40,41], indirectly affecting carbon exchange. Our results indicated that in 2023, there was no significant difference in NEE between moderate and severe salinization compared to mild salinization, whereas in 2024, NEE was significantly reduced under both moderate and severe salinization conditions. In 2023, GEP showed no significant difference between moderate and severe salinization compared to mild salinization, while in 2024, GEP decreased significantly under both moderate and severe salinization. Interannual precipitation modulates plant growth and photosynthetic capacity by altering soil moisture and salinization conditions, thereby further regulating the effect intensity and direction of carbon flux [12]. Wet years with abundant precipitation promote plant growth and facilitate photosynthesis and respiration, thus resulting in a small influence of salinization on ecosystem carbon exchange. Conversely, dry years may exacerbate salinization stress, creating unfavorable soil conditions for plant growth, and thus significant variations in carbon exchange occur across different salinization gradients. Our study revealed significant seasonal dynamics in carbon exchange, with significantly greater differences in NEE, ER, and GEP under extreme conditions in the middle of the growing season (June–August) than at the beginning (May) and end (September) of the growing season. During the early growing season, groundwater-mediated capillary rise transports soil salts to the surface, inducing topsoil salinization [42], which suppresses plant growth and eliminates significant variations in NEE, ER, and GEP across saline–alkaline gradients. During the mid-growing season, elevated soil moisture establishes more favorable hydrological conditions for plant development [43], consequently enhancing carbon fluxes. Under extreme salinization conditions, the survival environment of plants further deteriorates; plants have to make more drastic adjustments to their biomass allocation to sustain themselves, and this eventually stops ecosystem carbon exchange from increasing in parallel with hydrothermal conditions. During the late growing season, foliar senescence becomes prevalent across all saline–alkaline gradients, as evidenced by widespread leaf yellowing and substantial declines in physiological activity [9], and this results in no difference in carbon fluxes between saline–alkaline grasslands. Our study further identified significant positive correlations between both AGB and BGB with NEE, ER, and GEP under salinization conditions (Figure 7). These relationships emerge because saline–alkaline conditions impair plant growth and reduce litter accumulation—consequently affecting both gross ecosystem productivity (GEP) and heterotrophic respiration [26,31]. Results showed that biomass declined significantly under extreme salinization, and that increased salinization led to the gradual disappearance of poorly saline-tolerant species and a shift in community dominance towards saline-tolerant species, resulting in a simplification of species composition [23]. When salinization increased, redundancy analysis showed that R/S had the highest explanatory power for carbon exchange among various factors (Figure 8). To adapt to adverse conditions, plants have adjusted their biomass allocation strategies: they typically increase below-ground biomass inputs to enhance water and nutrient uptake, while reducing above-ground fractions to minimize transpiration losses. Additionally, plants increase the proportion of biomass allocated to roots [20], and this shift in plant biomass allocation indirectly decreases GEP by modifying leaf photosynthetic activity. In our study, the average CUE over two years was significantly lower under extreme salinization due to the disproportionate response of carbon fluxes: stress caused greater synergistic declines in GEP and NEE relative to ER [12]. With the intensification of saline–alkaline stress, soil nutrient levels decline markedly [24], which impairs the carbon sequestration capacity of grassland ecosystems [11,44]. Additionally, as an integrated indicator of carbon cycle processes, CUE dynamics are strongly limited by the combined synergistic effects of temperature and precipitation [45,46]. In summary, ST, SM, and biomass allocation collectively regulate ecosystem carbon exchange processes.

Results showed that the interaction between biotic and soil factors was the primary driver behind the significant differences in carbon exchange patterns observed between extreme saline–alkaline grasslands and their moderate-to-severe saline–alkaline counterparts. AGB and BGB exhibited progressive declines with increasing salinization, though this declining trend was significantly attenuated within the soil pH range of 9.0–10.0. Saline–alkaline stress triggers adaptive reorganization of plant communities, which is characterized by the enhanced dominance of alkaline-tolerant species that offset the decline in alkaline-sensitive plants, thus preserving biomass stability [23]. Concurrently, elevated soil moisture in saline–alkaline soils enhances water availability for these tolerant species [38]. This enhancement of hydrological conditions supports the synchronous growth of above-ground and below-ground plant organs, while activating root-related photosynthetic and respiratory functions, which in turn modulates ER and GEP [9]. However, when saline–alkaline stress exceeded plant tolerance thresholds, grassland ecosystems exhibited rapid declines in NEE, ER, and GEP (Figure 4). Under extreme conditions, plants exhibit adaptive responses to the deteriorating soil environment by significantly increasing their below-ground biomass allocation, a shift that results in elevated R/S [15,33], while beneficial for maintaining basic plant physiological functions, this ultimately reduces overall ecosystem carbon exchange capacity [11]. On a global scale, increasing above-ground biomass via reseeding or nutrient management plays a crucial role in grassland carbon sequestration and the restoration of degraded saline–alkaline grasslands [11,13]. In summary, carbon exchange exhibits distinct responses along the saline–alkaline gradients: while maintaining comparable levels between moderate and severe saline–alkaline grasslands through the compensatory effects of soil moisture and biomass allocation, it shows significant reduction under extreme conditions.

## 4. Materials and Methods

### 4.1. Study Site

The study was conducted at the Shanxi Youyu Loess Plateau Grassland Ecosystem National Observation and Research Station, located in Youyu, Shanxi Province (112°19′ E, 39°59′ N, altitude 1348 m). The experimental area belongs to a temperate continental monsoon climate, with an annual average temperature of 4.6 °C (maximum monthly average 20.2 °C, minimum monthly average −14.0 °C) and a frost-free period of approximately 100–120 days. The average annual precipitation is 425 mm (1991–2018), mostly concentrated in July and August, accounting for over 60% of annual rainfall. The soil type is chestnut soil (Chinese classification). The dominant species is *Leymus secalinus* (Georgi) Tzvel., with associated species including *Saussurea amara* (L.) DC. (meadow saussurea) and *Artemisia anethifolia* Weber. The soil pH of the topsoil is from 8.0–10.7, and the bulk density of the 0–5 cm surface soil is 1.34 ± 0.03 g cm^−3^.

### 4.2. Experimental Design

Representative salinized grassland sites were selected and divided into different patches based on soil pH values to represent grasslands with different salinization degrees. Four soil pH gradients were identified, specifically mild salinization (pH = 8.36 ± 0.01), moderate salinization (pH = 9.21 ± 0.06), severe salinization (pH = 9.92 ± 0.04), and extreme salinization (pH = 10.49 ± 0.01). The experimental plots were designed to be 6 m × 6 m in area, with a 10 m distance between plots, and a total of 24 plots were set. The samples were arranged in an area of approximately 300 m × 40 m.

### 4.3. Measurement of Ecosystem Carbon Fluxes

Metal bases (0.5 m × 0.5 cm × 0.1 m) were driven into grassland plots. The ecosystem carbon fluxes of the grassland ecosystem during the growing season was measured between 8:00 and 12:00, twice a month. We conducted five measurements during the peak growth period in 2023 (to avoid the impact of the metal bases hitting the soil in the early growing season, we only conducted monitoring in the middle of the growing season) and ten measurements throughout the entire growing season in 2024. Carbon fluxes were measured using a gas analyzer (LI–840, LI–COR, Lincoln, NE, USA) connected to a transparent assimilation chamber (0.5 m × 0.5 m × 0.5 m), a filter, and a gas pump. During NEE measurement, the assimilation chamber was positioned on the metal base to ensure a tight seal. Data were recorded using the gas analyzer software (LI–840, Version 2.0.1), with a set time of 80 s. Following NEE measurement, the assimilation chamber was covered with an opaque hood and ventilated. After the internal CO_2_ concentration stabilized to ambient levels, the chamber was resealed on the base to measure ER. The data recording and measurement time were the same as those for the NEE. Finally, GEP was calculated using formula [12]. CO_2_ fluxes are calculated using Equation (1):(1)$$Fc=\frac{VP_{av}(1000-W_{av})}{RS(T_{av}+273)}\times \frac{dc}{dt}$$

Among them, Fc was the ecosystem CO_2_ flux (μmol·m^−2^·s^−1^); V was the chamber volume (m^3^), i.e., V = length × width of the assimilation chamber × (chamber height + exposed height of the base); Pav was the average atmospheric pressure in the chamber during measurement (kPa); Wav was the water vapor partial pressure in the chamber during measurement (mmol·mol^−1^); R was the universal gas constant (8.314 J·mol^−1^·K^−1^); S was the area of the assimilation chamber (m^2^); Tav was the average temperature in the chamber during measurement (°C); and dc/dt was the slope of CO_2_ concentration change over the measurement period. The differences between the NEE and ER were defined as the GEP. Among them, NEE is the net ecosystem fluxes of CO_2_ between the ecosystem and the atmosphere. ER is the rate of CO_2_ emission from both plant and soil respiration. GEP is the gross amount of CO_2_ fixed by vegetation through photosynthesis per unit time.

The WUE was calculated using the ER, GEP, and ET via Equation (2), whereas the CUE was then calculated using ER and GEP via Equation (3). Data for ER, GEP, and ET were obtained from measurements with the gas analyzer (LI–840, LI–COR, Lincoln, NE, USA).WUE = (GEP − ER)/ET(2)CUE = (GEP − ER)/GEP(3)

### 4.4. Measurement of Plant Biomass and Soil Physicochemical Properties

Above-ground biomass (AGB) was sampled in mid-August 2024 at the seasonal peak. All plants vegetation and litter within a randomly placed 0.2 m × 1 m quadrat were harvested at ground level. After collection, plant samples were placed in a forced-air oven at 105 °C for 30 min to terminate moisture and then oven-dried at 65 °C until a constant weight was achieved. The dry weight was measured and scaled to grams per square meter to represent AGB. Following above-ground harvest, soil cores (7 cm diameter) were extracted from four depth layers (0–10, 10–20, 20–30, and 30–40 cm) within the same plots. Each layer was bagged in a mesh root bag, rinsed with water to isolate roots, and manually cleaned to remove debris. The below-ground biomass (BGB) was determined after oven-drying the roots at 65 °C to a constant weight. Finally, the root-to-shoot ratio (R/S) was calculated as BGB/AGB.

Soil temperature (ST, 0–10 cm) and soil moisture (SM, 0–10 cm) were measured at the same time as ecosystem carbon fluxes. Soil temperature was measured using a digital thermometer, while soil moisture was measured using a portable soil moisture tachymeter (TDR–350, Spectrum Technologies, Inc., Aurora, IL, USA). Soil pH values were determined with a ratio of soil to the water of 1:2.5 (*w*/*v*); then, the soil was shaken for 30 min, and a pH meter was used to determined soil pH values.

### 4.5. Statistical Analysis

Repeated-measurement ANOVA was used to test the effects of different degrees of salinization and year on GEP, ER, NEE, SM, and ST. One-way ANOVA was used to test the differences of different degrees of salinization on the GEP, ER, NEE, SM, ST, WUE, CUE, AGB, BGB, and R/S, and Duncan’s method was used for multiple comparisons. One-way ANOVA was also conducted to test the significant difference of GEP, ER, NEE, SM, and ST among treatments in each measurements during two growing seasons. The correlations between ecosystem carbon fluxes and SM, ST, soil pH, AGB, BGB, and the R/S were evaluated using linear or non-linear regression, according to the higher coefficient of determination (R^2^). The correlations between the two-year average of ecosystem carbon fluxes and variables were measured in 2024: soil pH, AGB, BGB, and the R/S. Furthermore, redundancy analysis (RDA) was employed to explore the overall relationships between ecosystem carbon fluxes and soil physicochemical properties, ecosystem resource use efficiency, plant biomass, and allocation. SPSS 25 (SPSS Inc., Chicago, IL, USA) software was used for data analysis, and Origin 2021 (OriginLab Corporation, Northampton, MA, USA) and Canoco 5.0 (Microcomputer Power, Ithaca, NY, USA) software were used for plotting.

## 5. Conclusions

Our research presented the general pattern of ecosystem carbon fluxes in grassland under different levels of salinization. We found that the ecosystem carbon fluxes declined significantly under extreme salinization, while no significant difference was observed between moderate and severe saline–alkaline grasslands. The ecosystem carbon fluxes were jointly regulated by plants, R/S, soil temperature, soil moisture, and soil pH, among which the R/S exhibited the greatest explanation to the variation in ecosystem carbon fluxes. In this study, we highlight that the distribution of plant biomass was identified as a key factor influencing the carbon sequestration potential in saline–alkaline grasslands. These findings have significant implications for the restoration of degraded saline–alkaline grasslands and for enhancing grassland carbon sequestration capacity.

## Figures and Tables

**Figure 1 plants-14-02854-f001:**
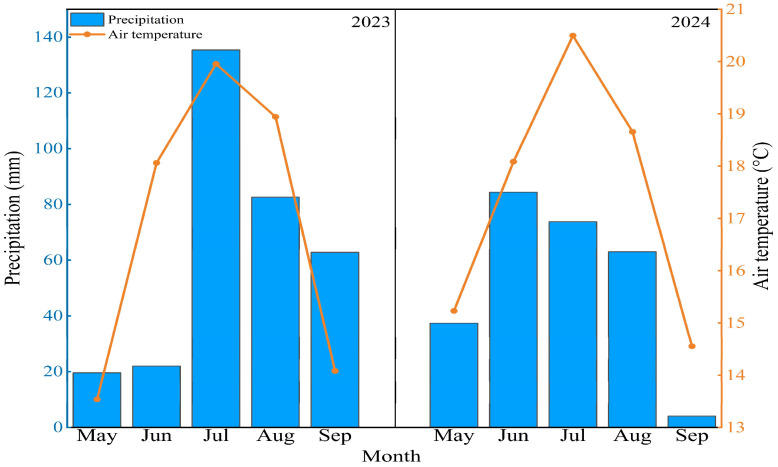
Monthly precipitation and average air temperature during growing seasons in 2023 and 2024.

**Figure 2 plants-14-02854-f002:**
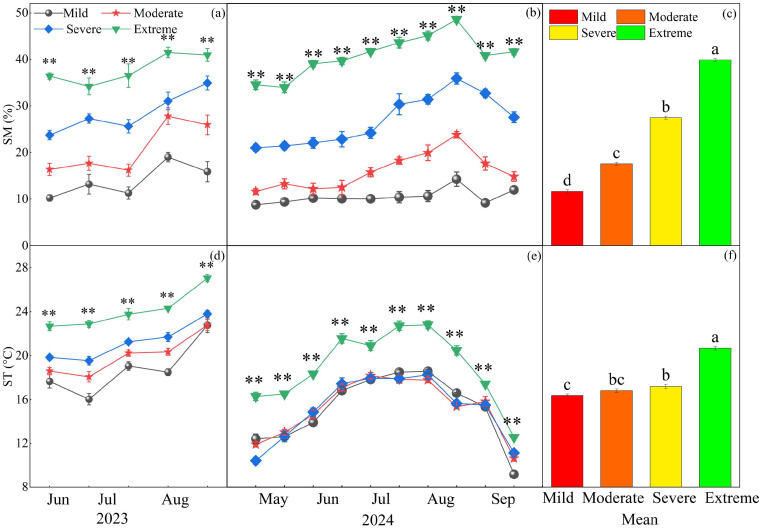
Effect of different gradients of salinization on soil temperature (ST) and soil moisture (SM). (**a**) SM in 2023; (**b**) SM in 2024; (**c**) the two-year average of SM; (**d**) ST in 2023; (**e**) ST in 2024; (**f**) the two-year average of ST. ‘**’ indicates a significant difference among the four gradients of salinization at the same measurement time point (*p* < 0.01), calculated based on the average values of each treatment replicate. Different letters denote significant differences among treatments (*p* < 0.05).

**Figure 3 plants-14-02854-f003:**
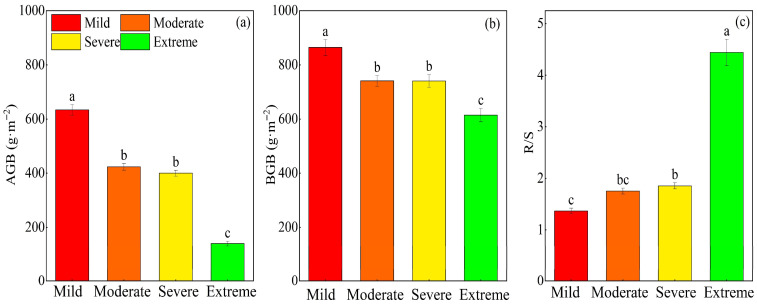
Effect of different gradients of salinization on above-ground biomass (AGB), below-ground biomass (BGB), and root-to-shoot ratio (R/S). (**a**) The average of AGB; (**b**) the average of BGB; (**c**) the average of R/S. Different letters denote significant differences among treatments (*p* < 0.05).

**Figure 4 plants-14-02854-f004:**
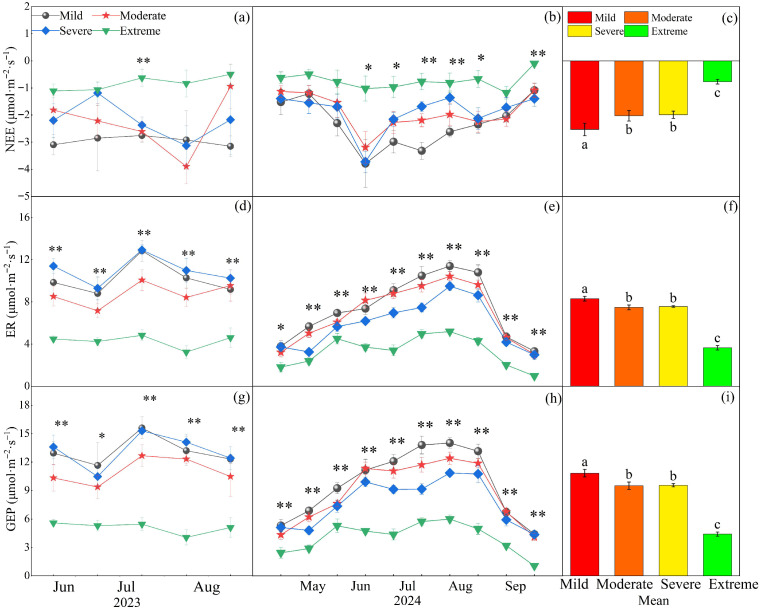
Effects of different gradients of salinization on net ecosystem carbon exchange (NEE), ecosystem respiration (ER), and gross ecosystem productivity (GEP). (**a**) NEE in 2023; (**b**) NEE in 2024; (**c**) the two-year average of NEE; (**d**) ER in 2023; (**e**) ER in 2024; (**f**) the two-year average of ER; (**g**) GEP in 2023; (**h**) GEP in 2024; (**i**) the two-year average of GEP. ‘*’ and ‘**’ represent statistical significance at *p* < 0.05 and *p* < 0.01, respectively, indicating whether there is a significant difference among the four salinization gradients in each measurement based on one-way ANOVA analysis. Different letters denote significant differences among treatments (*p* < 0.05).

**Figure 5 plants-14-02854-f005:**
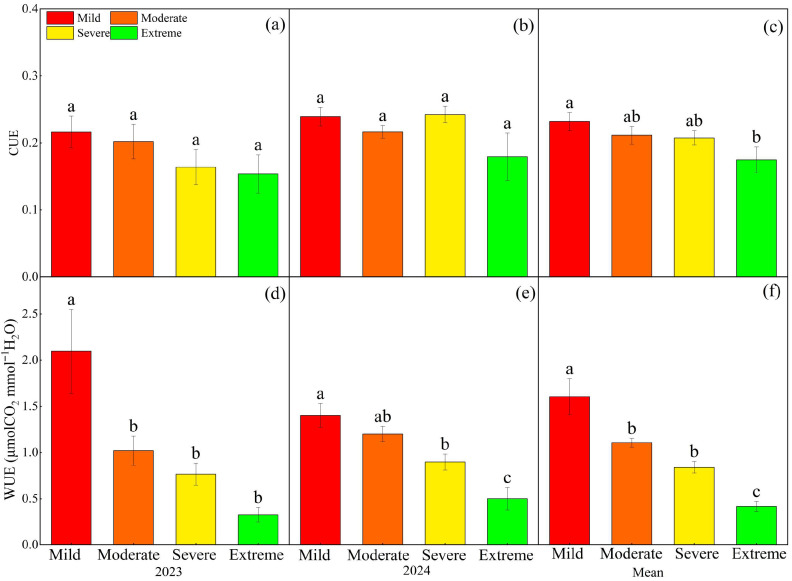
Effects of different gradients of salinization on ecosystem carbon use efficiency (CUE) and ecosystem water use efficiency (WUE) in 2023–2024. (**a**) CUE in 2023; (**b**) CUE in 2024; (**c**) the two-year average of CUE; (**d**) WUE in 2023; (**e**) WUE in 2024; (**f**) the two-year average of WUE. Different letters denote significant differences among treatments (*p* < 0.05).

**Figure 6 plants-14-02854-f006:**
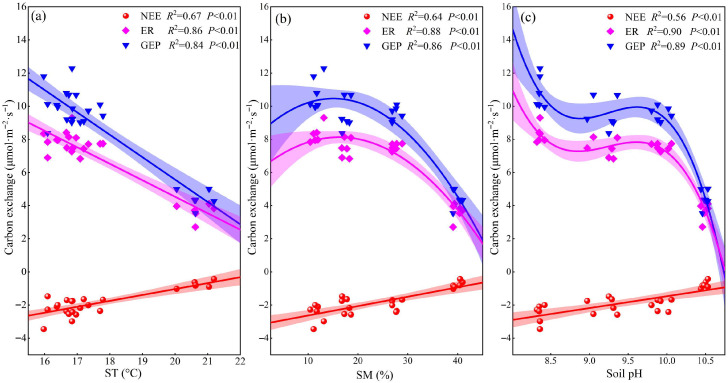
Correlation of ecosystem carbon fluxes with soil temperature (ST), soil moisture (SM) and soil pH. (**a**) Correlation of ecosystem carbon fluxes and ST; (**b**) correlation of ecosystem carbon fluxes and SM; (**c**) correlation of ecosystem carbon fluxes and soil pH. NEE: net ecosystem carbon exchange; ER: ecosystem respiration; GEP: gross ecosystem productivity.

**Figure 7 plants-14-02854-f007:**
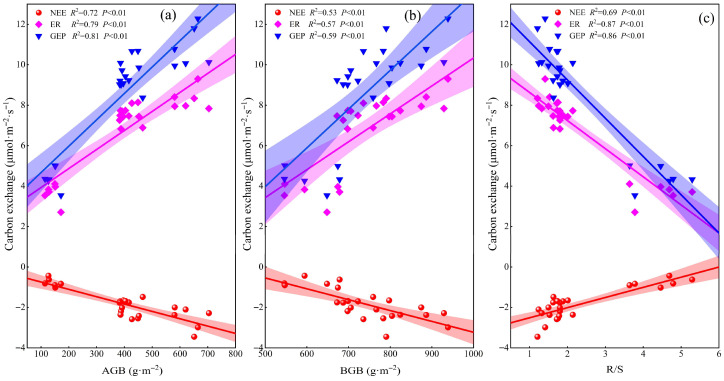
Correlation of ecosystem carbon fluxes with above-ground biomass (AGB), below-ground biomass (BGB) and root-to-shoot ratio (R/S). (**a**) Correlation of ecosystem carbon fluxes and AGB; (**b**) correlation of ecosystem carbon fluxes and BGB; (**c**) correlation of ecosystem carbon fluxes and R/S. NEE: net ecosystem carbon exchange; ER: ecosystem respiration; GEP: gross ecosystem productivity.

**Figure 8 plants-14-02854-f008:**
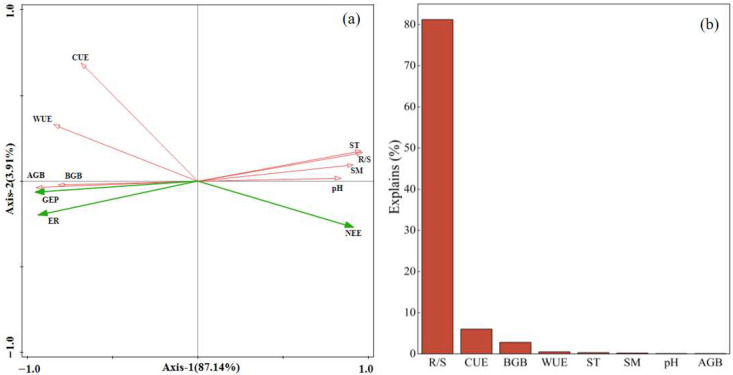
Redundancy analysis of ecosystem carbon fluxes with soil factors; biomass and its allocation and resource use efficiency; and relative contribution of factors. (**a**) Redundancy analysis; (**b**) the explanatory power of various factors. NEE: net ecosystem carbon exchange; ER: ecosystem respiration; GEP: gross ecosystem productivity; ST: soil temperature; SM: soil moisture; AGB: above-ground biomass; BGB: below-ground biomass; R/S: root-to-shoot ratio; CUE: ecosystem carbon use efficiency; WUE: ecosystem water use efficiency.

**Table 1 plants-14-02854-t001:** Results (F values) of repeated-measures ANOVA testing the effects of year and salinization on NEE, ER, GEP, CUE and WUE.

	NEE	ER	GEP	CUE	WUE	ST	SM
Year	6.73 *	93.90 **	82.41 **	4.55 *	0.15	267.72 **	10.11 **
Salinization	31.60 **	67.97 **	82.52 **	2.40	16.85 **	45.92 **	483.79 **
Year × Salinization	0.65	10.25 **	7.71 **	0.77	2.47	2.81 *	9.89 **

Note: ‘*’ and ‘**’ represent *p* < 0.05 and *p* < 0.01, respectively. NEE: net ecosystem carbon exchange; ER: ecosystem respiration; GEP: gross ecosystem productivity; CUE: ecosystem carbon use efficiency; WUE: ecosystem water use efficiency; ST: soil temperature; SM: soil moisture.

## Data Availability

Data will be made available on request.

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
