# Peer review of "Seasonal Dynamics of Ecosystem Carbon Exchange and Their Influencing Factors in Grasslands of Different Degrees of Salinization in Northern China"

_plants, 2025, doi:10.3390/plants14182854_

Round 1

Reviewer 1 Report

Comments and Suggestions for Authors

This manuscript investigates the seasonal carbon exchange dynamics of saline-alkaline grasslands and identifies the root-to-shoot ratio (R/S) as a key regulatory factor. Although the study addresses a topic of significant ecological importance, major revisions are required to establish scientific rigor. The specific suggestions are as follows:

  1. Inconsistent use of “salinization” and “saline-alkaline” was observed throughout the manuscript. To ensure clarity, “saline-alkaline stress” (a compound stressor of salinity and alkalinity) should be used uniformly, replacing “salinization” wherever applicable.
  2. Line 44: The term “carbon exchange” lacks definition upon first mention. To address this, the phrase “carbon exchange processes (i.e., net ecosystem exchange, NEE; ecosystem respiration, ER; gross ecosystem productivity, GEP)” should be added at its initial occurrence for clarity.
  3. The introduction section primarily lists previous conclusions and lacks a focus on research gaps. For example: (Line 50-51) “Previous studies have mainly concentrated on... whereas research on saline-alkaline grasslands has been relatively limited”. However, it does not explicitly identify specific limitations. Additionally, (Line 82-87) the transition from the literature review to the research objectives is abrupt, and it does not clarify how this study addresses the research gaps. Suggestion: Organize the logic as follows: Effects of saline-alkaline on soil/plants→ Mechanisms of carbon exchange responses → Limitations of existing research. Please add some transitional sentences before presenting the research question.
  4. Line 97-99: The positive correlation between SM and saline-alkaline is contradictory, and the authors should explain this.
  5. Line 127-135: Results section lacks key finding: “No significant difference in carbon exchange between moderate/severe saline-alkaline conditions” lacks emphasis.
  6. Line 165: The core role of R/S is mentioned in just one sentence, without forming a logical chain with the aforementioned biomass data.
  7. Line 122: “The 2023 ER/GEP peak is in July.” It is inappropriate to use Figure 3 directly. Recommendation to cite specific figure 3a.
  8. The results section is logically confusing. It is recommended that the logical framework of the results section be reorganized.
  9. The discussion section merely repeats the result that “extreme saline-alkaline significantly inhibits carbon exchange” without explaining why there is no difference between moderate and severe saline-alkaline at pH 9.0-10.0, nor does it correlate this with the data from this study.
  10. Line 274-281: The core role of R/S has not been fully substantiated. It merely states that “R/S has the highest explanatory power.”
  11. Please delete irrelevant content:

Line 228-236: The general effect of temperature on photosynthesis is unrelated to the theme of saline-alkaline.

Line 307-308: Low saline-alkaline stress adaptability exceeds the gradient range of this manuscript's research.

  1. Please use present tense for established facts; past tense for specific study findings. Address similar issues: Line 37: “has been showing” change to “shows”; Line 60-61etc.
Comments on the Quality of English Language

I noticed a few syntax errors within the text, which would need to be addressed.

Reviewer 2 Report

Comments and Suggestions for Authors

The overall quality of the manuscript is good to me. However, some technical writing could be strengthened for better readability.

  1. Abstract & Introduction section. The novelty of the study should be emphasized. The cited literature does provide the background of this research, but the research gap was not clearly mentioned. The two research questions (lines 88–90) are not clearly phrased. 
  2. Method section. The measurement frequency is unequal in the years 2023 (5) and 2024 (10). Should justify why different numbers of measurements were taken each year. (line 336-337)
  3. Figures.  I noticed those small letters (“a”, “b”, “c”, “d”) above the bars in Figure 1c and 1f (also in some other figures) are results of post-hoc multiple comparison tests (in this case Duncan’s test, as described in the Methods section), right? I didn't find any discussion of that info. in the writings or figure captions. It's better to discuss this data or add some descriptions in the figure caption, so reader, if they are unfamiliar with those notions, will understand what it means. Redundancy analysis (Figure 7) could benefit from a clearer text value on each bar, as most bars are too small/low to read.
  4. Discussion section. I would say most of this part is of good quality. Some issues/suggestions as follows: many explanations are restated in different ways (e.g., high soil pH → reduced microbial activity → reduced respiration is mentioned multiple times). This makes the discussion long but not deeper without synthesis, but rather repetition. Broader implications: the discussion remains very site- and condition-specific (Shanxi saline-alkaline grasslands) but does not place the results strongly in a broader global carbon cycle or grassland degradation framework. Highlights mechanistic drivers (R/S, pH, SM) clearly. Ends with why this matters — both for theory and for management/policy.

Reviewer 3 Report

Comments and Suggestions for Authors

In this study, four groups of saline-alkali gradients were classified based on soil pH values: mild, moderate, severe and extreme. This gradient design is reasonable, with the pH value range covering the typical degree of salinization (8.36-10.49), and six repeated sample plots are set for each group, ensuring sufficient sample size. The authors comprehensively determined three core carbon flux parameters: NEE (net ecosystem carbon exchange), ER (ecosystem respiration), and GEP (gross ecosystem productivity). And it was associated with key factors such as soil temperature, humidity and biomass. It is particularly interesting to note that they take the root-to-crown ratio (R/S) as a key regulatory factor. This study systematically reveals the dynamic laws of carbon exchange in grasslands under saline-base gradients and the core driving role of R/S, which has significant reference value for the restoration of degraded grasslands and carbon management. Overall, this manuscript has the potential for publication, but the following aspects need to be noted.

  1. In 2023, there were only 5 samplings compared to 10 samplings in 2024 (L 336-337). It is necessary to further explain the impact of this difference on the seasonal dynamic analysis. For example, whether it has been processed through standardization.
  2. Soil moisture (SM) was quadratic correlated with ER/GEP (L 167-169), but the specific value of the "critical threshold" was not quantified. For instance, what are the SM thresholds for inhibiting ER and GEP respectively? This point is interesting and requires in-depth discussion.
  3. The discussion mentioned that microbial activities were inhibited (L 200-203), but no measured microbial biomass/community data were obtained, weakening the chain of evidence for microbe-driven carbon cycling.
  4. This study is based on data from 2023 to 2024, but does not discuss the potential impact of interannual climate fluctuations (such as precipitation differences) on the results. The authors mentioned interannual differences in the discussion section but did not conduct an in-depth analysis (L 248-252). Would this affect the universality of the conclusion?
  5. The authors can compare the differences in carbon sink potential between natural grasslands and saline-alkali grasslands. Although this part was mentioned in the introduction (L 50-54), it was not deepened in the discussion.
  6. This study lacks data on soil organic carbon (SOC) storage, making it difficult to assess the impact of carbon exchange changes on the stability of the carbon pool. In the conclusion section (L 398-402), the authors can look forward to future research based on this point.
  7. It is strongly recommended that authors carefully examine their manuscripts to avoid any detail errors (for example, L 406). At the same time, do not have very long paragraphs, as this makes it difficult for readers to understand (for example, L 226-290).

Reviewer 4 Report

Comments and Suggestions for Authors

Dear Authors, 

I want to ask you to improve your manuscript as suggested below.

There is a huge difference in the data for the mentioned months in 2023 and 2024. I must be uniform or should have fewer fluctuations. 

Figure 1: The statistical ** meaning is missing. One (*) and its meaning is also meaning. 

Why are statistical ** given with only extreme salinization levels? Why are these symbols not provided on other levels, like mild, moderate, and severe?

It will be better to mention the meaning of and for significant details in captions. 

Figure 2: The details of the figures' captions related to statistical letters are missing. 

Table 1: The full form of the abbreviation of traits is missing. 

Figure 3: The statistical notations of * and ** are missing.

The given data seems to be insufficient to prove the conclusion of this manuscript, even with the correlation analysis.   

Round 2

Reviewer 1 Report

Comments and Suggestions for Authors

The authors have made effective and careful revisions, which have significantly improved the quality of the manuscript.

The changes address the key concerns raised during the review process, and the clarity, structure, and overall presentation of the work have been greatly enhanced. 

Author Response

Thank you for taking the time to review the manuscript amidst your busy schedule. Wishing you success in your work and all the best.

Reviewer 3 Report

Comments and Suggestions for Authors

The authors carefully revised the manuscript, and the revised version is now acceptable.

Author Response

Thank you for your recognition of the manuscript. I wish you good health and success in your work.

Reviewer 4 Report

Comments and Suggestions for Authors

Dear authors, 

Thank you for revising your manuscript and providing your response. However, your response is not justifiable regarding the data of 2023, only 3 months (approximately 90 days), while in 2024, the data is given for 5 months (approximately 150 days). This is a huge difference; however, try to provide a reason in full detail (either wet or dry year, or any other reason) in the methodology section, and short details in the abstract section. Also discuss this reason in the discussion, along with a citation of related work. 

Figure 2 and Figure 4 still have (*) and (**) on one type of treatment. Why not these statistical notations are provided on other treatment lines? It will be better to provide these symbols with all treatment lines, or you need to mention a statement (The significance symbols (*) and (**) denote statistical significance, calculated from the monthly average values of the treatments) in the captions.  
